# Identification of Long Noncoding RNAs That Exert Transcriptional Regulation by Forming RNA–DNA Triplexes in Prostate Cancer

**DOI:** 10.3390/ijms24032035

**Published:** 2023-01-19

**Authors:** Yugang Liang, Yali Lu, Qin Chen, Yihang Cheng, Yunsheng Ma, Yan Huang, Minyan Qiu, Yao Li

**Affiliations:** State Key Laboratory of Genetic Engineering, Shanghai Engineering Research Center of Industrial Microorganisms, School of Life Science, Fudan University, Shanghai 200433, China

**Keywords:** RNA–DNA triplex, transcriptional regulation, biomarker, prostate cancer, lncRNA–AD000684.2

## Abstract

Long noncoding RNAs (lncRNAs) are involved in transcriptional regulation, and their deregulation is associated with the development of various human cancers, including prostate cancer (PCa). However, their underlying mechanisms remain unclear. In this study, lncRNAs that interact with DNA and regulate mRNA transcription in PCa were screened and identified to promote PCa development. First, 4195 protein-coding genes (PCGs, mRNAs) were obtained from the The Cancer Genome Atlas (TCGA) database, in which 1148 lncRNAs were differentially expressed in PCa. Then, 44,270 pairs of co-expression relationships were calculated between 612 lncRNAs and 2742 mRNAs, of which 42,596 (96%) were positively correlated. Among the 612 lncRNAs, 392 had the potential to interact with the promoter region to form DNA:DNA:RNA triplexes, from which lncRNA AD000684.2(AC002128.1) was selected for further validation. AC002128.1 was highly expressed in PCa. Furthermore, AD000684.2 positively regulated the expression of the correlated genes. In addition, AD000684.2 formed RNA–DNA triplexes with the promoter region of the regulated genes. Functional assays also demonstrated that lncRNA AD000684.2 promotes cell proliferation and motility, as well as inhibits apoptosis, in PCa cell lines. The results suggest that AD000684.2 could positively regulate the transcription of target genes via triplex structures and serve as a candidate prognostic biomarker and target for new therapies in human PCa.

## 1. Introduction

Prostate cancer (PCa) is one of the most common cancers in men and a leading cause of cancer-related deaths worldwide [1,2]. The molecular mechanisms that control the tumorigenesis and progression of PCa remain unclear. Long noncoding RNAs (lncRNAs) affect various cell phenotypes (e.g., proliferation, apoptosis, cell cycle, invasion, and metastasis), suggesting that they have specific regulatory functions in cancers [3,4]. Certain lncRNAs are differentially expressed in PCa and play either inhibitory or promoting roles in tumors [5]. They are thought to act through their interaction with DNA, RNA, and proteins in the cell. LncRNA–miRNA and lncRNA–protein interactions have been reported extensively [6,7]; however, limited studies have focused on the interaction of lncRNAs with DNA.

Studies confirmed that lncRNAs interact with DNA and/or proteins and act as transcriptional activators or repressors to regulate gene expression [8,9]. lncRNAs can directly bind to purine-rich double-stranded DNA sequences through Hoogsteen base pairing to form an RNA–DNA triplex [10]. Considering the difference in the base composition of the RNA strand, classical triplexes are divided into three categories, namely, pyrimidine motifs (Y), purine motifs (R), and mixed motifs (M) [11,12]. When pyrimidine-rich RNA strands form triplexes with DNA, the protonation of cytosine requires acidic pH, which is rare in biological systems. Therefore, under biological conditions, triplexes are more likely to be formed by purine-rich RNA and DNA double strands. Bioinformatic analysis has predicted a large number of triplex-forming motifs in the entire genome. Interestingly, these motifs tend to accumulate in gene regulatory regions, particularly in the promoter region [13,14]. Triplexes may have important regulatory roles based on bioinformatics analysis and preliminary experimentation; however, in-depth experimental evidence is lacking.

The purpose of this paper is to identify potential lncRNAs that interact with DNA in the promoter region by analyzing differentially expressed lncRNAs in prostate cancer that perform their biological functions and promote prostate cancer development by interacting with DNA to form triple-stranded bodies.

In the present study, computational analysis predicted that 392 lncRNAs can interact with the promoter region to form triplexes in PCa. Among them, the unstudied lncRNA AD000684.2 (AC002128.1) was selected for further validation. AD000684.2 was highly expressed in PCa tissues and positively regulated the expression of positively correlated genes. Further research on the regulatory mechanism revealed that AD000684.2 formed RNA–DNA triplexes with the promoter region of the regulated gene through *trans* action. Therefore, lncRNA AD000684.2 promotes cell proliferation and motility and inhibits apoptosis in PCa cell lines.

## 2. Results

### 2.1. Differential Expression Profiles in PCa Samples

EdgeR was used to perform differential gene expression analysis of RNA-seq data from 498 PCa tissues and 52 normal tissues from the TCGA database. According to our screening criteria of differential genes as |log FC| < log 21.5 and FDR < 0.05, 5946 genes with differential expression in cancer and normal tissues were obtained (Figure 1A), of which 3174 genes were upregulated, and 2772 genes were downregulated in cancer tissues (Appendix A). These genes include 4195 protein-coding genes (PCGs, mRNAs), 1148 lncRNAs, and 630 other genes.

GO and KEGG pathway analyses were used to gain some insight into the cellular processes of dysregulated in PCa, and the data revealed that the differentially expressed genes were involved in protein transport, cell division, and signal transduction (Figure 1B–D). Results show that the development of PCa may be related to the connection of cells to the extracellular matrix, exchange of substances between cells and the outside world, and signaling (Figure 1E).

The association of differentially expressed genes with long-term overall survival (OS) and long-term disease-free survival (DFS) were assessed in PCa. A one-way COX proportional risk regression model was used to calculate the hazard ratio (HR) of each gene in the model, in which the expression of differentially expressed genes was considered as the independent variable, and the corresponding *p*-value was calculated. An HR greater than one indicates that the gene is deleterious, and the risk ratio increases with the increase in gene expression. An HR less than one indicates that the gene is favorable, and the risk ratio decreases with the increase in gene expression. Among all the differentially expressed genes in PCa, 84 genes were associated with OS, and 175 genes were associated with DFS (Table 1).

The patients were divided into high- and low-expression groups according to the expression of OS genes and DFS genes in patients, and Kaplan–Meier survival analysis was performed on the prostate according to the expression differences of each gene. The results showed a significant difference (*p* < 0.05) in the survival rate between the high- and low-expression groups for the OS (Appendix A) and DFS genes (Appendix A). OS and DFS genes remarkably affected the OS and DFS among patients with PCa.

Among the 84 OS genes, 65 mRNAs and 14 lncRNAs were identified. In total, 63 genes had an HR greater than 1, 21 genes had an HR less than 1, 65 genes had a log FC greater than 0, and 19 genes had a log FC less than 0. The distributions of these genes are presented in Table 1.

Among the 175 DFS genes, 163 mRNAs and 8 lncRNAs were identified. A total of 170 genes had an HR greater than 1, 5 genes had an HR less than 1, 166 genes had a log FC greater than 0, and 9 genes had a log FC less than 0. The distributions of these genes are presented in Table 1.

As shown in Table 1, most of the differentially expressed genes in PCa shortened the OS and DFS (HR > 1), and the expression of these unfavorable genes was mostly elevated in PCa (log FC > 0). A small percentage of differentially expressed genes prolonged OS and DFS (HR < 1), but the expression of these favorable genes was mostly decreased in PCa (log FC < 0). This result suggests that the differentially expressed genes associated with OS and DFS in PCa showed a trend of high expression of deleterious genes and low expression of favorable genes.

### 2.2. Correlation Analysis for mRNAs and lncRNA Genes Using Gene Expression Data

Co-expression analysis was carried out between 1148 lncRNAs and 4195 mRNAs by calculating the Pearson correlation coefficients of gene expression. At |*r*| ≥ 0.6 and FDR < 0.01, between the lncRNA and mRNA, a co-expression relationship was considered to exist between the lncRNA and mRNA. Overall, 44,270 pairs of co-expression relationships were found between 612 lncRNAs and 2742 mRNAs, of which 42,596 (96%) were positively correlated and 1314 were negatively correlated (Appendix A). In addition, among the 44,270 pairs, the maximum negative r value of negatively correlated gene pairs was −0.76, while the positive correlation gene pairs with the *r*-value exceeding 0.76 had 4532 pairs with the highest *r*-value of 0.98.

Furthermore, among the 612 lncRNAs, 532 (87%) were 100% positively correlated with mRNA, and 605 (99%) were positively correlated with ≥50% mRNAs (Figure 1F). According to the analysis of the subcellular localization of these lncRNAs using the lncATLAS database (lncATLAS; crg.eu), 73.4% of the 612 lncRNAs showed negative RCI (Relative Concentration Index), and 77.8% of the 393 lncRNAs showed negative RCI, indicating that they may be located in the nucleus (Appendix A).

### 2.3. Prediction of Potential Triplex-Forming lncRNAs and Triplex Structure in the Promoter Region of Co-Expression Genes

Considering that most of these co-expression pairs were positively correlated and that most of these lncRNAs were localized in the nucleus, we speculated that one of the reasons for co-expression pairs was the regulation of mRNA expression by lncRNAs at the transcriptional level. Generally, lncRNAs promote or repress transcription by interacting with transcription-related proteins. The interaction between RNA and DNA to form triplexes can also participate in transcription regulation. In the present study, we focused on predicting the triplex-forming ability of 612 lncRNAs with DNA-interacting potential.

Triplex analysis was performed using Triplexator analysis [14,15]. The complete list of coordinates of PTS for these lncRNA is given in Appendix A.

Among these 612 lncRNAs, 392 lncRNAs could interact with the promoter region to form triplex-forming oligos (TFOs) (Appendix A). The distribution of the 392 lncRNAs in different proportions of positively correlated mRNAs was similar to that of the 612 lncRNAs and did not change significantly. LncRNAs, which were positively correlated with >50% of mRNAs, accounted for 97.70% (Figure 1G).

In total, 29,731 PTS were identified in the promoter sequences of 1508 mRNAs and 392 lncRNAs, including 7839 purine motifs (R), 12,678 pyrimidine motifs (Y), and 9214 purine–pyrimidine motifs (mixed/M; Table 2, Appendix A). A total of 392 lncRNAs and 1508 mRNAs are listed in Appendix A, and the top 20 lncRNAs with the most interactions are shown in Table 3.

Among the 392 potential triplex-forming lncRNAs, 5 lncRNAs (HOTAIR, NEAT1, FENDRR, PARTICLE, and KCNQ1OT1) were reported as experimentally validated triplex-forming lncRNAs [16,17,18], and a few of them (such as MALAT1) were predicted by others without experimental validation [19].

### 2.4. AD000684.2 and ASMTL-AS1 Are Highly Expressed in PCa

Nine candidate lncRNAs were selected for further consideration based on the ranking of PTS scores (Appendix A) and on several other parameters of lncRNAs, including logFC (log of fold change) of lncRNA expression, RCI, Log rank *p* of DFS, number of co-expression pairs, and percentage of positively correlated mRNA, where the number of co-expression pairs is the highest priority parameter. The results are shown in Appendix A. AD000684.2 and ASMTL-AS1, which had 16 and 31 potential tri-plex target mRNAs, were selected for further study (Appendix A). The analysis of the expression levels of AD000684.2 and ASMTL-AS1 in PCa from the TCGA database revealed that AD000684.2 and ASMTL-AS1 were highly expressed in PCa tissues (Figure 2A), and the expression was more significant in tumors with higher malignancy (Figure 2B). Kaplan–Meier analysis showed that compared with patients with high AD000684.2 or ASMTL-AS1 expression patients with PCa having low AD000684.2 or ASMTL-AS1 expression had longer DFS (Figure 2C). In combination with the analysis of the lncATLAS database, the results show that AD000684.2 and ASMTL-AS1 were mainly located in the nuclei of various cancer cell lines (Figure 2D). Therefore, AD000684.2 and ASMTL-AS1 are likely to regulate gene expression at the transcriptional level.

### 2.5. AD000684.2 Positively Regulates the Expression of Correlated Genes

To investigate whether AD000684.2 and ASMTL-AS1, which are highly expressed in PCa, are involved in regulating the expression of positively correlated genes, we selected five genes with high correlation coefficients with AD000684.2 and ASMTL-AS1 for validation. Two siRNAs that specifically target AD000684.2 and ASMTL-AS1 were used to decrease their expression, and RT-PCR was carried out to detect the expression levels of the positively correlated genes corresponding to these two lncRNAs. Results show that the siRNA-mediated silencing of AD000684.2 significantly downregulated the expression levels of these five positively correlated genes in C4-2B cells (Figure 3A). However, after knocking down the expression of ASMTL-AS1, the expression of its positively correlated genes did not decrease significantly (Figure 3B). Therefore, AD000684.2 positively regulates the expression of positively correlated genes.

According to the analysis of the expression of AD000684.2 positively regulated genes (SLC23A3, CPT1B, MSH5, KAT2A, and LYG1) of PCa in the TCGA database, these five genes were highly expressed in PCa tissue (Figure 3C), and survival curve analysis showed that these five genes had a certain prognostic value (Figure 3D).

### 2.6. AD000684.2 Forms RNA–DNA Triplexes with the Promoter Region of Regulated Genes

The above results show that lncRNA–AD000684.2 could positively regulate gene expression at the transcriptional level. Therefore, AD000684.2 could directly bind to GA-rich DNA sequences through Hoogsteen base pairing to form RNA–DNA triplexes and be anchored in gene transcriptional regulatory regions, possibly by promoting the formation of gene transcription initiation complexes to regulate gene expression. As shown in Table 4, possible TFOs of AD000684.2 were predicted. Next, the sequence 5 kb upstream of the gene transcription start site (TSS) was obtained, the potential TTSs that may form an RNA–DNA triplex with AD000684.2 were predicted, and the triplex motifs of AD000684.2 with the promoters of target genes are listed in Table 5. The highest scoring TFO sites were concentrated upstream of the AD000684.2 transcription start site (+484/+528).

An electrophoretic mobility shift assay (EMSA) was used to verify the triplex formation of AD000684.2 with gene regulatory regions. The AD000684.2 RNA fragment containing the TFO site was constructed by in vitro transcription, and a carboxyfluorescein (FAM)-tagged DNA fragment containing the TTS site in the promoter region of these regulated genes was synthesized. Next, the RNA fragment and FAM-labeled DNA were incubated in a buffer, and triplex formation was monitored on a 12% polyacrylamide gel. The results show that the number of DNA–RNA triplexes increased with increasing RNA molar concentration. By contrast, when the TFO site of the RNA fragment was mutated (AD000684.2-mut), triplex complex formation was not observed (Figure 4). These results provide strong evidence that AD000684.2 forms RNA–DNA triplexes with the promoter region of the regulated gene.

### 2.7. AD000684.2 Promotes Cell Proliferation and Motility, as Well as Inhibits Cell Apoptosis, in PCa Cell Lines

The biological functions of AD000684.2 in the occurrence and development of PCa were investigated by testing the effects on cell proliferation, migration, and apoptosis after knocking down AD000684.2 in PCa cell lines. The cell proliferation assay showed that the cells transfected with AD000684.2 siRNA had a decreased proliferative capacity relative to cells transfected with the control oligos in C4-2B and PC-3 cells (Figure 5A). Furthermore, transwell migration and wound healing assays showed that the knockdown of AD000684.2 expression inhibited the motility of PCa cells compared with the control (Figure 5B,C). Next, the role of AD000684.2 in apoptosis in PCa was explored. C4-2B and PC-3 cells were treated with si-AD000684.2 or nonspecific siRNA control, stained with Annexin V–FITC/PI, and analyzed by flow cytometry. The results showed that the knockdown of AD000684.2 cells increased the percentage of early and late apoptotic cells (Figure 5D). These results indicate that AD000684.2 promotes cell proliferation and motility and inhibits cell apoptosis in PCa cell lines, possibly acting as a proto-oncogene in the development of PCa.

## 3. Discussion

Differentially expressed lncRNAs and mRNAs, especially genes associated with OS and with DFS, may be involved in cancer development and progression in some form. To explore the lncRNAs that play a role in PCa, we used RNA-seq data from prostate cancer samples in the TCGA database to screen for significantly differentially expressed lncRNAs and mRNAs and analyze them for prognosis-related genes. The differentially expressed RNAs we identified could serve as potential candidate diagnostic targets for PCa and deserve further investigation.

Considering that lncRNAs can regulate the expression of target mRNAs at the transcriptional level, we performed co-expression analysis between differentially expressed lncRNAs and mRNAs. Among these co-expression pairs, some may be caused by LncRNAs transcriptionally regulating the expression of target genes. In this paper, we focus on lncRNAs that can interact with DNA to form RNA–DNA triplexes.

LncRNAs positively or negatively regulate the transcription of target genes by interacting with DNA to form triplexes, mainly because the triplex structure alters the interaction of proteins, such as transcription factors or epigenetic regulators, with DNA in the promoter region, thus affecting transcription. More negatively regulated lncRNAs have been reported than positively regulated lncRNAs. For example, Phillip reported that lncRNA Fendrr acts via dsDNA/RNA triplex formation at target regulatory elements and directly increases PRC2 occupancy at these sites [16]. ANRASSF1 forms an RNA–DNA hybrid and recruits PRC2 to the RASSF1A promoter, and then PRC2 induces the accumulation of the repressor marker H3K27me3, thereby reducing the transcriptional activity of RASSF1A [20]. The lncRNA REG1CP forms an RNA–DNA triplex with a high purine extension on the distal promoter of the REG3A gene. The DNA helicase FANCJ is connected to the core promoter of REG3A through a triplex, releases double-stranded DNA, and promotes the glucocorticoid receptor α (GRα)-mediated transcription of REG3A [21]. Moreover, noncoding transcripts from the minor promoter of the dfhr (~400 bp) gene repress the transcription of downstream protein coding genes by forming a purine–purine–pyrimidine triplex motif with the DHFR promoter [22,23].

In the present study, most of the co-expression pairs of lncRNAs and mRNAs were positively correlated. Our previous study revealed that the RNA–DNA triplex structure of lncRNA–AP006284.1 can increase chromatin accessibility and promote transcription, allowing it to act as an enhancer (https://www.sciencedirect.com/science/article/pii/S2352304222001039, accessed on 27 April 2022). Jalali et al. analyzed the expression correlation of lncRNAs and their computational targets by virtue of lncRNA binding to the promoter of the target gene; only 23 lncRNAs showed a positive correlation with 51 genes, while none showed a significant negative correlation [15]. This finding is consistent with our results.

On the basis of the above analysis, we hypothesized that lncRNAs increase chromatin accessibility by interacting with DNA in the genome to form a triplex structure, thus promoting transcription, which acts mainly in a *trans* manner. However, transcriptional regulation is complex, and the triplex structure alters the binding of epigenetic regulators and transcription factors to the promoter region. Therefore, it exhibits different regulatory patterns. In addition, lncRNA alone can recruit proteins to the promoter region, thus affecting transcription. Therefore, an in-depth experimental validation of each lncRNA is required.

In the present study, AD000684.2 was subjected to further investigation. AD000684.2 could form an RNA–DNA triplex structure in the promoter region of positively regulated genes in a *trans* manner, and the EMSA assay was performed to verify the formation of triplexes.

LncRNAs are important in the initiation and progression of various cancers, including PCa [4,24]. In the present study, lncRNA AD000684.2 was highly expressed in PCa, and AD000684.2 expression was significantly correlated with the prognosis of patients with PCa. The results of the cell phenotype experiment showed that AD000684.2 promotes cell proliferation and motility and inhibits cell apoptosis in PCa cell lines, possibly acting as a proto-oncogene in the development of PCa. We also found that genes positively regulated by AD000684.2 (SLC23A3, CPT1B, MSH5, KAT2A, and LYG1) were highly expressed in PCa tissues and had certain prognostic value.

In summary, this study demonstrated that AD000684.2 was upregulated in PCa and positively regulated the expression of positively correlated genes by forming RNA–DNA triplexes. Moreover, AD000684.2, which acts as an oncogene, could be used for PCa diagnosis and prognosis prediction. However, further studies are required to define the detailed mechanisms of AD000684.2 in PCa treatment.

## 4. Materials and Methods

### 4.1. Computational Analysis

#### 4.1.1. RNA-Seq Data Acquisition of PCa

We collected RNA sequencing data from 498 primary prostate cancer tissue samples and 52 normal prostate tissue samples from the Genomic Data Commons (GDC) data portal of The Cancer Genome Atlas (TCGA, https://portal.gdc.cancer.gov, accessed on 10 August 2018) and then screened for differentially expressed genes according to our previously reported approach [25].

RNA-seq data included count and FPKM data. Count data refer to the number of sequencing fragments contained in each gene. FPKM data refer to the number of fragments per million mappings per thousand bases of each gene, which is generally used to measure gene expression. Two forms of gene expression were used in this study: logFPKM and standardized logFPKM (zFPKM):$$\mathrm{logFPKM}=\log_{2}\mathrm{FPKM}$$
$$\mathrm{zFPKM}=\frac{\mathrm{logFPKM}-\mathrm{mean}\left(\mathrm{logFPKM}\right)}{\mathrm{sd}\left(\mathrm{logFPKM}\right)}$$

#### 4.1.2. Significantly Differential Gene Expression Analysis

For the RNA-seq data, EdgeR was used for differential gene expression analysis. The TMM algorithm [26] was used for standardized data processing, combined with a negative binomial distribution generalized linear model and quasi-likelihood estimation [27,28,29], to calculate the differential expression of each gene in cancer and normal tissues, including log FC and *p*-values; log FC is the logarithm of the expression of multiple genes in cancer tissues compared to that in normal tissues. The Benjamini–Hochberg algorithm [30] was used to correct the *p*-value and obtain the FDR value (multiple hypothesis testing correction was conducted to obtain FDR value).

Gene Ontology (GO) function enrichment analysis was performed in three functional ontologies: biological process, cellular component, and molecular function [31]. Kyoto Encyclopedia of Genes and Genomes (KEGG) pathway enrichment analysis was also performed to identify pathways enriched in PCa using MAS3.0 system (http://bioinfo.capitalbio.com/mas3/, accessed on 21 March 2019). The *p*-value was calculated by hypergeometric distribution, and a pathway with *p* < 0.05 was considered significant.

#### 4.1.3. COX Proportional Hazards Regression Analysis

The same expression data for detection of differentially expressed genes and the clinical data from TCGA were used for COX proportional hazards regression analysis. The R language package survival was used to calculate the COX proportional hazards model.

#### 4.1.4. Pearson Correlation Coefficient between Differentially Expressed lncRNAs and mRNAs in PCa

Using the zFPKM data, the Pearson correlation coefficient between lncRNAs and mRNAs was calculated to represent the correlation between the two genes [32]. The correlation calculated using this method is called the expression correlation, indicated as $r_{\mathrm{XY}}$: $$r_{\mathrm{XY}}=\frac{\mathrm{cov}\left(X,\;Y\right)}{\delta_{X}\delta_{Y}}=\frac{\sum_{i=1}^{n}\left(X-\bar{X}\right)\left(Y-\bar{Y}\right)}{\sqrt{\sum_{i=1}^{n}{\left(X-\bar{X}\right)}^{2}}\sqrt{\sum_{i=1}^{n}{\left(Y-\bar{Y}\right)}^{2}}}$$
where X and Y represent the expression levels of lncRNA and mRNA in different samples.

#### 4.1.5. Acquisition of the lncRNA Sequences and Promoter Sequences of Genes

We obtained the 5 kb promoter sequence upstream of the transcription start site (TSS) of PCa-related genes using R software, HOMER software, and the sequence of PCa-related lncRNAs through GENCODE (https://www.gencodegenes.org/human/, accessed on 16 November 2021). The sequences of the promoters and lncRNAs were obtained to predict RNA–DNA triplexes.

#### 4.1.6. In Silico Prediction of RNA–DNA Triplexes

The potential DNA:DNA:RNA triplex sites between lncRNAs and the promoter sequences of the correlated mRNAs were predicted using Triplexator (http://bioinformatics.org.au/tools/triplexator/manual.html, accessed on 20 December 2021), which is a computational framework for the in silico prediction of triplex structures. The parameters for predicting the PTSs of lncRNA were as follows: Triplexator -l 17 -e 20 -m R,M -fr on -mrl 7 -mrp 1 -of 0 -po -o ENSG00000281404.tfo -ss ENSG00000281404.fasta. The TFO predictions were performed with the parameters -l 17 -e 20 -g 20 -m R/Y/M -fm 0 -of 1 -fr of (where l is the lower length bound, e is the error rate less than 10%, g is the guanine rate less than 20%, m is the type of triplex motif; R is a purine motif, Y is a pyrimidine motif, M is a purine–pyrimidine motif, fr denotes filter repeats, and of is the output format).

### 4.2. Experimental Validation

#### 4.2.1. In Vitro Transcription

We constructed the DNA fragments using the primer sequences listed in Appendix A. First, 1 μg of the DNA fragment containing T7 promoter as the template was added to 50 mM DTT, 2 mM biotin NTP mix, 20 U of RNase inhibitor, 50 U of T7 RNA polymerase, and transcription buffer. After mixing, the mixture was incubated at 37 °C for 2 h. Next, 20 U of RNase-free DNase I was added at 37 °C for 15 min to digest the template DNA. Next, 100 μL of chloroform was added to extract RNA.

#### 4.2.2. Electrophoretic Mobility Shift Assay (EMSA)

First, 1 pmol of 5′FAM-labeled DNA fragments and excess AD000684.2 RNA fragments were incubated in a buffer containing 40 mM Tris-acetate (pH 7.5), 20 mM KCl, 10 mM Mg(CH_3_COO)_2_, and 10% glycerol at 37 °C for 1 h. It was then incubated with 0.5 U of RNase H or with 0.5 ng of RNase A for 30 min at room temperature. The formation of triplexes was monitored by 12% polyacrylamide gel electrophoresis. The FAM-labeled DNA sequences are presented in Appendix A.

#### 4.2.3. Cell Culture

Human prostate cancer cell lines C4-2B and PC-3 were obtained from the Shanghai Cell Bank of the Chinese Academy of Sciences. PC-3 cells were cultured in RPMI-1640 medium supplemented with 10% fetal bovine serum (FBS). C4-2B cells were cultured in DMEM supplemented with 10% FBS. All cells were cultured at 37 °C in a 5% CO_2_ incubator.

#### 4.2.4. RNA Interference and Transfection

All small interfering RNA (siRNA) oligonucleotides were purchased from GenePharma (Shanghai, China) and were used at a concentration of 100 pmol. Transfection was performed using the Hilymax Transfection Reagent (Dojindo Laboratories, Kumamoto, Japan) according to the manufacturer’s instructions. Knockdown efficiency was tested using RT-qPCR. The siRNA sequences are listed in Appendix A, and the primer sequences are listed in Appendix A.

#### 4.2.5. Cell Proliferation Assay

Cell proliferation assays were performed as described previously using the Cell Counting Kit-8 (Dojindo Laboratories, Kumamoto, Japan) according to the manufacturer’s instructions. Absorbance was measured at 450 nm using a microplate reader ELx808 (Biotek, Winooski, VT, USA) at different time points. Absorbance at 615 nm was used as a reference.

#### 4.2.6. Annexin V–FITC Apoptosis Assay

Apoptosis was determined using an FITC–Annexin V Apoptosis Detection Kit (Dojindo Laboratories, Kumamoto, Japan). The green fluorescent label Annexin V–FITC and the red fluorescent label PI were used to distinguish early and late apoptotic cells from dead cells. After staining for 15 min in the dark, an FACSCalibur flow cytometer (BD Biosciences, Franklin Lakes, NJ, USA) was used for detection.

#### 4.2.7. Transwell Migration Assay

Cells (10^5^ cells/100 μL) were seeded in the upper chamber of a 24-well transwell (353097, Corning, NY, USA), and 700 μL of medium containing 10% FBS was added to the lower chamber. After incubation for 24 h, the cells were fixed with 4% paraformaldehyde for 15 min, and the upper membrane surface cells were removed with a cotton swab, stained with 0.1% crystal violet, and counted in five random fields using an optical microscope (Olympus, Tokyo, Japan).

#### 4.2.8. Wound Healing Assay

The cells were maintained in six-well plates and transfected with siRNA. After 4 h of transfection, cells were washed with 1× PBS. A 1 mL pipette tip was used to make three smooth and vertical lines in the hole. After streaking, the cells were washed with 1× PBS, serum-free medium was added, and images were captured under a microscope at 0 h. The six-well plate was placed back into the incubator for culturing, removed after incubation for 24 h, and imaged under a microscope.

#### 4.2.9. Statistical Analysis

The numerical data were presented as the mean ± standard deviation (SD) of at least three determinations, which were analyzed using GraphPad Prism 8 software. Statistical comparisons between groups of normalized data were conducted using a *t*-test, according to the test conditions. Statistical significance was set at *p* < 0.05, with a 95% confidence interval.

## Figures and Tables

**Figure 1 ijms-24-02035-f001:**
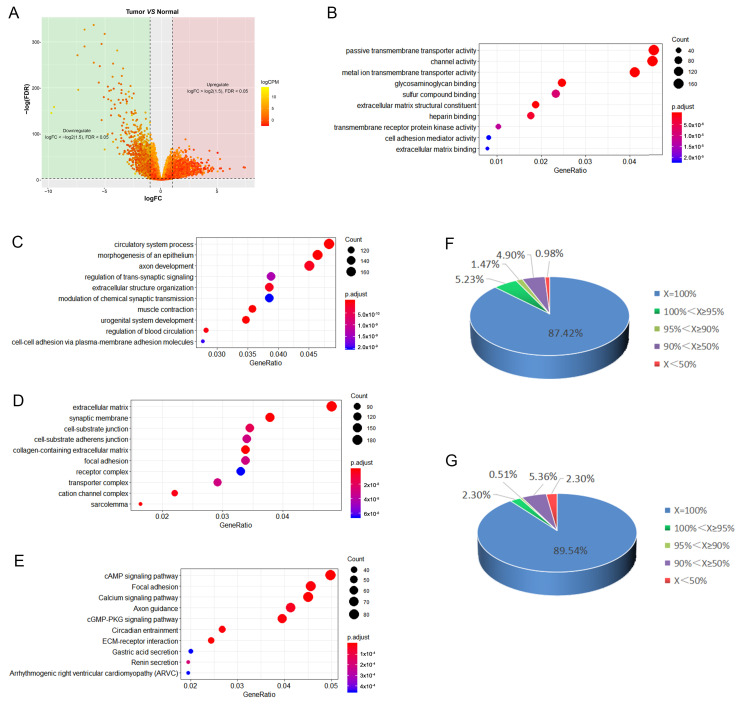
(**A**) Volcano plot of differential gene expression analysis in PCa compared with normal tissue (fold change ≥ 1.5; *p*-value < 0.05); (**B**) GO biological process, molecular function enrichment analysis; (**C**) GO biological process, biological process enrichment analysis; (**D**) GO biological process, cellular component enrichment analysis; (**E**) KEGG pathway enrichment analysis. The gene ratio in the figure indicates the ratio of the number of differentially expressed genes containing the annotation to the number of differentially expressed genes; (**F**) proportion of lncRNAs across the different percentages of positively correlated mRNA; distribution of 612 lncRNAs across the different percentages of positively correlated mRNA. X = number of positively correlated mRNA/total number of correlated mRNAX100%; (**G**) proportion of lncRNAs across the different percentages of positively correlated mRNA; distribution of 392 lncRNAs acting as potential third strand of potential triplex sites (PTSs) across the different percentages of positively correlated mRNA; X = number of positively correlated mRNA/total number of correlated mRNA × 100%.

**Figure 2 ijms-24-02035-f002:**
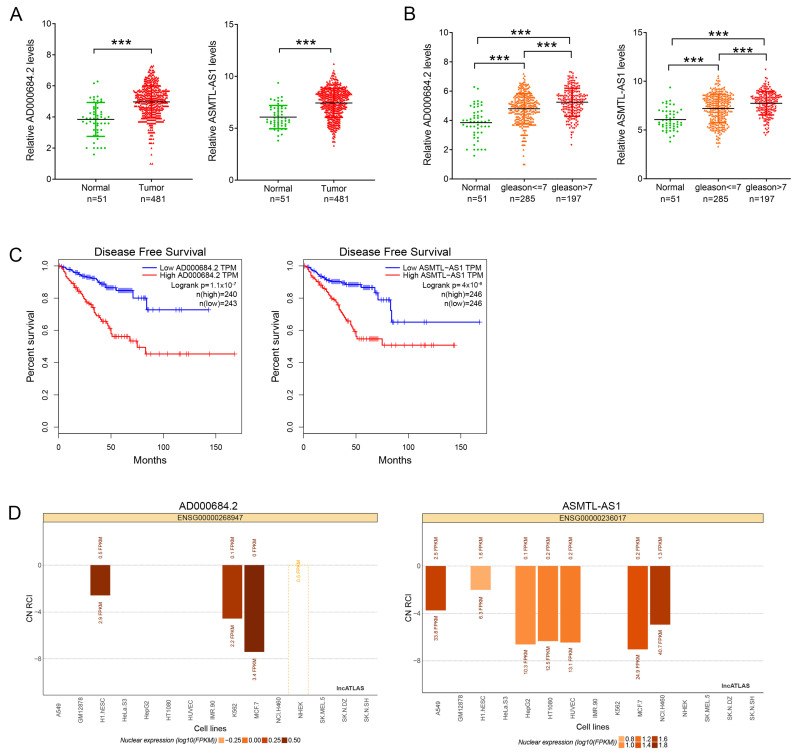
AD000684.2 and ASMTL-AS1 are highly expressed in PCa: (**A**) analysis of the expression levels of AD000684.2 and ASMTL-AS1 in PCa tissues and normal tissues acquired from the TCGA database (*** *p* < 0.001); (**B**) analysis of the relationship between the expression of AD000684.2 and ASMTL-AS1 and the pathological grade of PCa; (**C**) Kaplan–Meier curves for disease-free survival of PCa patients in a published dataset acquired from TCGA using the high quartile AD000684.2 or ASMTL-AS1 levels as the cutoff; (**D**) subcellular localization of AD000684.2 and ASMTL-AS1 in various cell lines obtained from the LncATLAS database.

**Figure 3 ijms-24-02035-f003:**
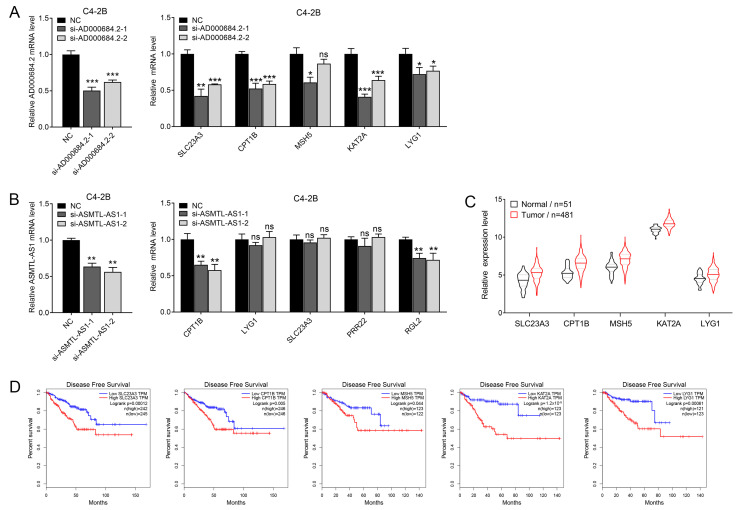
AD000684.2 positively regulates the expression of positively correlated genes: (**A**) C4-2B cells were transfected with siRNA to specifically knock down the expression of AD000684.2, and then the mRNA levels of AD000684.2 and positively related genes were simultaneously detected by RT-qPCR (*n* = 3; data are presented as the mean ± SD; * *p* < 0.05; ** *p* < 0.01; *** *p* < 0.001; ns, not significant); (**B**) the siRNAs specifically targeting ASMTL-AS1 were transfected into C4-2B cells, and then RT–qPCR was used to simultaneously detect the mRNA levels of ASMTL-AS1 and positively correlated genes (*n* = 3; data are presented as the mean ± SD; ** *p* < 0.01; ns, not significant); (**C**) analysis of the expression levels of SLC23A3, CPT1B, MSH5, KAT2A, and LYG1 in PCa tissues and normal tissues in the TCGA database; (**D**) Kaplan–Meier curves of disease-free survival (RFS) in PCa patients with high or low expression of SLC23A3, CPT1B, MSH5, KAT2A, and LYG1.

**Figure 4 ijms-24-02035-f004:**
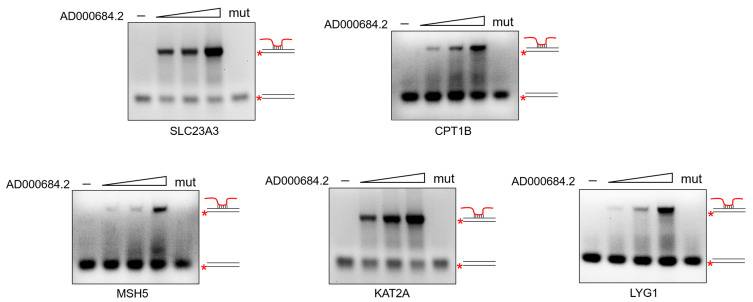
AD000684.2 forms RNA–DNA triplexes with the promoter region of regulated gene. The 0.5 pmol double-stranded 5′FAM-labeled oligonucleotide comprising TTS with a molar excess (40-, 80-, and 160-fold) of AD000684.2 RNA fragment (+367/+576) or 80-fold AD000684.2-TFO-mut fragment was incubated at 37 °C for 1 h, and the formation of RNA–DNA triplexes was monitored by EMSA. * (FAM)-tagged DNA fragment.

**Figure 5 ijms-24-02035-f005:**
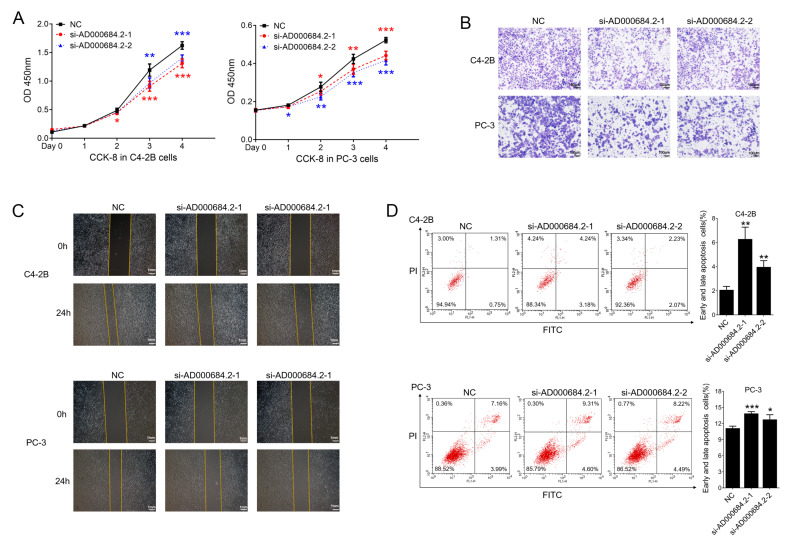
AD000684.2 promotes cell proliferation and motility and inhibits cell apoptosis in PCa cell lines: (**A**) cell proliferation analysis was performed with CCK-8 assay in C4-2B and PC-3 cells. Cells transfected with si-AD000684.2 and siNC were seeded into a 96-well plate at 5000 cells/well and examined at timepoints of 0, 1, 2, 3, and 4 days; (**B**) cell migration assay was performed in C4-2B and PC-3 cells. Cells transfected with si-AD000684.2 and siNC were seeded into a transwell chamber and then stained with 0.1% crystal violet after incubation for 24 h, bar = 100 μm; (**C**) wound healing assays were performed in C4-2B and PC-3 cells Cells were transfected with si-AD000684.2 and siNC, and scratch healing was observed after 24 h, bar = 100 mm; (**D**) cell apoptosis assay was performed in C4-2B and PC-3 cells. Cells were transfected with si-AD000684.2 and siNC, and then the percentage of early and late apoptotic cells was quantitated by Annexin V/PI staining using FACSCalibur flow cytometry (**A**–**D**, *n* = 3; data are presented as the mean ± SD; * *p* < 0.05; ** *p* < 0.01; *** *p* < 0.001; ns, not significant).

**Table 1 ijms-24-02035-t001:** The data of OS and DFS in terms of log FC and HR.

	OS ^1^		DFS ^1^	
	log FC > 0	log FC < 0	log FC > 0	log FC < 0
HR > 1	60	3	166	4
HR < 1	5	16	0	5

^1^ Overall survival (OS); disease-free survival (DFS).

**Table 2 ijms-24-02035-t002:** Statistics of the triplex motif PTS sites and TFOs in our study.

Motif Type	Number of Potential Triplex-Forming lncRNAs	Number of Potential Triplex Target mRNAs	Number of Potential Triplex-Forming Sites (PTSs)
R (purine motif)	236	1116	7839
Example: NDP	TFO ^1^: 3′–GAGAGAcAGAGAGAGAGAGAGAGAGGGAG–5′		
	||||||*||||||*|||||||||||||||		
	TTS ^1^: 5′–GAGAGAGAGAGAGGGAGAGAGAGAGGGAG–3′		
	3′–CTCTCTCTCTCTCCCTCTCTCTCTCCCTC–5′		
Y (pyrimidine motif)	195	1169	12,678
Example: CAV1	TFO: 3′–TTTGTGTGTGTGTTTTTTTGTGTGTGT–5′		
	||||||||||||||||||*|||||||||||		
	TTS: 3′–AGAGACAGAGACAGAAACAGAGAGAGA–5′		
	5′–TCTCTGTCTCTGTCTTTGTCTCTCTCT–3′		
M (purine–pyrimidine or mixed motif)	188	938	9214
Example: COL6A1	TFO: 3′–TTTGTGTGTGTGTTTTTTTGTGTGTGT–5′		
	|*|||*|||||*|*|||*|||||||||		
	TTS: 3′–AGAGACAGAGACAGAAACAGAGAGAGA–5′		
	5′–TCTCTGTCTCTGTCTTTGTCTCTCTCT–3′		
Total	619 ^2^	3223 ^3^	29,731

^1^ TFO triplex-forming oligos (lncRNA sequence); TTS triplex target sites (duplex DNA sequence); | complementary base pair; * noncomplementary base pair; ^2^ after removing the duplicate lncRNA, the total number was 392; ^3^ after removing the duplicate mRNA, the total number was 1508.

**Table 3 ijms-24-02035-t003:** The Top 20 lncRNAs with the maximum number of potential triplex target mRNAs.

Ranking	The lncRNA Symbol	lncRNA Ensemble	Number of Potential Triplex Target mRNAs	Percentage of Positively Correlated mRNAs ^1^
1	ADAMTS9-AS2	ENSG00000241684	692	98.55
2	MBNL1-AS1	ENSG00000229619	595	99.33
3	HCG11	ENSG00000228223	553	98.73
4	AC107959.1	ENSG00000245025	497	100.00
5	LINC00654	ENSG00000205181	390	98.72
6	AF111167.2	ENSG00000259319	390	100.00
7	LINC01679	ENSG00000237989	374	100.00
8	AC004846.1	ENSG00000258376	373	100.00
9	NR2F1-AS1	ENSG00000237187	328	100.00
10	BOLA3-AS1	ENSG00000225439	300	99.67
11	MEF2C-AS1	ENSG00000248309	298	100.00
12	LINC00900	ENSG00000246100	279	100.00
13	DNM3OS	ENSG00000230630	272	100.00
14	AP002884.1	ENSG00000250303	267	100.00
15	ACTA2-AS1	ENSG00000180139	227	100.00
16	AC080013.1	ENSG00000240207	224	100.00
17	LINC00857	ENSG00000237523	221	100.00
18	FENDRR	ENSG00000268388	215	100.00
19	CASC15	ENSG00000272168	175	100.00
20 ^2^	AF165147.1	ENSG00000232855	164	100.00

^1^ Calculation formula: X = number of positively correlated mRNA/total number of correlated mRNA × 100%; ^2^ number of target mRNA 392.

**Table 4 ijms-24-02035-t004:** TFOs predicted by Triplexator.

lncRNA Symbol	Start	End	TFOs (5′–3′) ^1^	Score ^2^
AD000684.2	464	482	AAAGGGGAAGtAAGcAAA	16
AD000684.2	498	528	AAGAGAAGAcAGAAGAcAGAGAGAGAGGGA	28
AD000684.2	496	526	GtAAGAGAAGAcAGAAGAcAGAGAGAGAGG	27
AD000684.2	495	525	GGtAAGAGAAGAcAGAAGAcAGAGAGAGAG	27
AD000684.2	494	524	AGGtAAGAGAAGAcAGAAGAcAGAGAGAGA	27
AD000684.2	493	523	AAGGtAAGAGAAGAcAGAAGAcAGAGAGAG	27
AD000684.2	492	522	GAAGGtAAGAGAAGAcAGAAGAcAGAGAGA	27
AD000684.2	491	521	GGAAGGtAAGAGAAGAcAGAAGAcAGAGAG	27
AD000684.2	490	520	GGGAAGGtAAGAGAAGAcAGAAGAcAGAGA	27
AD000684.2	489	519	GGGGAAGGtAAGAGAAGAcAGAAGAcAGAG	27
AD000684.2	488	518	AGGGGAAGGtAAGAGAAGAcAGAAGAcAGA	27
AD000684.2	486	516	GcAGGGGAAGGtAAGAGAAGAcAGAAGAcA	26
AD000684.2	484	514	AAGcAGGGGAAGGtAAGAGAAGAcAGAAGA	27

^1^ TFOs, triplex-forming oligos; ^2^ score, triplex-forming potential scores.

**Table 5 ijms-24-02035-t005:** Triplex motif of lncRNA–AD000684.2 (RNA TFO) with target genes (DNA duplex or TTS).

Triplex Motif	TFO Start	TFO End	Duplex Gene	TTS Start	TTS End	Score
TFO: 3′–AGAAGACAGAAGAGAATGGA–5′	494	514	*SLC23A3*	1244	1264	16
||||||*||*||||*|*|||						
TTS: 5′–AGAAGAGAGGAGAGGATGGA–3′						
3′–TCTTCTCTCCTCTCCTACCT–5′						
TFO: 5′–GACAGAGAGAGAGGG–3′	512	527	*CPT1B*	599	614	12
||*|||**|||||||						
TTS: 3′–GACAGACGGAGAGGG–5′						
5′–CTGTCTGCCTCTCCC–3′						
TFO: 5′–AGAAGACAGAGAGAGA–3′	508	524	*MSH5*	1234	1250	13
||*|||**|||||||						
TTS: 3′–AGGAGAAAGGGAGAGA–5′						
5′–TCCTCTTTCCCTCTCT–3′						
TFO: 5′–AGAAGACAGAGAGAG–3′	506	521	*KAT2A*	961	976	13
|||*||*||||||||						
TTS: 3′–AGAGGAGAGAGAGAG–5′						
5′–TCTCCTCTCTCTCTC–3′						
TFO: 3′–AGGGAGAGAGAGACAGAAG–5′	509	528	*LYG1*	2726	2742	16
||*|||||||||**|||||						
TTS: 5′–AGAGAGAGAGAGTGAGAAG–3′						
3′–TCTCTCTCTCTCACTCTTC–5′						

* noncomplementary base pair

## Data Availability

Data are contained within the Appendix A.

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
