# Peer review of "Identification of Long Noncoding RNAs That Exert Transcriptional Regulation by Forming RNA–DNA Triplexes in Prostate Cancer"

_ijms, 2023, doi:10.3390/ijms24032035_

Round 1

Reviewer 1 Report

In this article, the authors analyzed the function of lncRNA in the formation of RNA-DNA triplexes involved in PCa.

This article shows some limitations in the sections’ general organization.

In the introduction section there are some point where the references are poor or missing with conflicting information on the role of lncRNA AD000684.2.

The aims of the study are very poor specified and also reported not in the properly section.

Regarding the methods section, the authors could reorganize this section maybe better defining the differences between in vitro and in silico analyses.  In addition, in some of the methods paragraphs some criticism are found.

Regarding the results section, many criticisms in the general description and organization have be found. For example, the authors indicated conflictual data regarding the tissue samples analyzed (line 339 and 499); the images should be reorganized specifically in the captions. The results are generally unclear.

The discussion section is very poor including phrases that the authors could reorganize in the introduction section. In addition, some parts of the discussion section are reported in other sections.

Therefore, I do not recommend the manuscript for publication. The manuscript should be extensively revised before an eventual resubmission.

Author Response

Hi Reviewer,

   please check the reply as the attached file, the revised manuscript has been submitted to the "English editing" service, you should be able to see it in a few days, thank you for your review, please feel free to communicate with me if you have any questions, thank you!

Reviewer 2 Report

It is an interesting study, the authors depended on both in silico analyses of available datasets as well as many validation studies using cell lines. In this manuscript, the authors screened and identified lncRNAs that interact with DNA and regulate mRNA transcription with subsequent promotion of prostate cancer development. They also detected the differentially expressed lncRNAs in prostate cancer in comparison to normal prostate. They detected 392 lncRNA that have the potential to interact with the promoter region to form DNA: DNA: RNA triplexes. Among those, they selected AD000684.2 (AC002128.1) for further study (expression and survival analysis) and validation. They also studied its relationship with five selected genes and show that it may serve as a candidate prognostic biomarker and target for developing new therapies in human prostate cancer

However, the following comments should be addressed

Major comments: 

1.     The justification of studying the AD000684.2 is not strong.

Need to justify more, why you selected these two lncRNA as they are not in the top 20 (table 3) nor they have the highest interaction.

2.     The discussion did not include many points of your results e.g. the prognostic value of AD000684.2 (based on the findings in section 2.4 of the results). Moreover, the effect on proliferation, motility, and apoptosis.

3.     In correlation analysis (2.2 in the result), Based on Table S6, when I tried to repeat the filter you mentioned in the result section, I could not come out with the same numbers of even r values you mentioned in the text. Please clarify this.

4.     The supplementary files you attached need very careful revision. 

Minor comments:

1.     Language revision should be done for the manuscript. 

2.     Please write in the abbreviate terms in full when first mentioned in the text

e.g. Line 124 ïƒ  PTS

Line 232 TFOs

Line 239 EMSA

Line 241FAM

Line 340GDC

3.     Line 29: Positively regulation ïƒ  please correct

4.     Line 72: please start the supplementary file with No. 1

5.     Line 81: “The association of differentially expressed genes with overall survival (OS) and disease-free survival (DFS) were assessed in PCa.”

Did you use the same dataset used for detection of Differentially expressed genes (DEGs), please mention.

6.     Line 82: is it risk ratio or hazard ratio? Please correct.

7.     Line 88: is it long or short OS and DFS, please mention clearly in the text.

8.     Line 91: what are the total survival genes, I think you mean the OS genes. If not, please clarify.

9.     Line 92-93: “……in patients and performed Kaplan-Meier survival analysis on prostate was performed according to the expression differences of each gene.” Please review.

10.  Line 94: cannot open S1, S2, S3 folders

11.  Line 114: Please make sure to increase the resolution of all the pictures included in the manuscript especially the written words. Please review the spelling inside the figures e.g. Figure 1 B and E

12.  Line 142: Based on the analysis of the subcellular localization of these lncRNAs by using the lncATLAS database, most of these lncRNAs were located in the nucleus”

Did you check the cellular localization of 612 lncRNA, if yes, please do not just write "most of these lncRNA" BUT instead mention the exact percentage of number of nuclear ones.

13.  Line 143: Table S5: This file does not have any of the mentioned data.

14.  Line 147-148: “the formation of co-expression pairs is caused by the regulation of mRNA expression by lncRNAs at the transcriptional level”.

This is like very strong suggestion based only on the co-expression and cellular localization.

15.  Line 153: Triplexator analysis: Please add reference, it is the first time to be mentioned

16.  For all of your validation studies, how many times you did the experiments?

Author Response

(The authors gave the same response as above.)

Round 2

Reviewer 1 Report

The authors have modified the manuscript. Then I can accept the manuscript for the publication.

Author Response

Dear Reviewer,

     Please check the coverletter and revised manuscript.thank you for your review.Do not hesitate to contact us if you have any questions regarding the coverletter and revised manuscript.

Reviewer 2 Report

I would like to thank the authors for their effort to reply the comments. 

Please find my comments in the attached file.

Author Response

(The authors gave the same response as above.)

Round 3

Reviewer 2 Report

The authors adjusted all the major points.